# Text encoders bottleneck compositionality in contrastive vision-language models

**Amita Kamath**[1]      **Jack Hessel**[2]      **Kai-Wei Chang**[1]

[1] University of California, Los Angeles
[2] Allen Institute for AI

{kamatha, kwchang}@cs.ucla.edu, jackh@allenai.org

## Abstract

Performant vision-language (VL) models like CLIP represent captions using a single vector. How much information about language is lost in this bottleneck? We first curate COMPPROMPTS, a set of increasingly compositional image captions that VL models *should* be able to capture (e.g., single object, to object+property, to multiple interacting objects). Then, we train *text-only recovery probes* that aim to reconstruct captions from single-vector text representations produced by several VL models. This approach does not require images, allowing us to test on a broader range of scenes compared to prior work. We find that: 1) CLIP's text encoder falls short on more compositional inputs, including object relationships, attribute-object association, counting, and negations; 2) some text encoders work significantly better than others; and 3) text-only recovery performance predicts multimodal matching performance on CONTROLLEDIMCAPS: a new evaluation benchmark we collect and release consisting of fine-grained compositional images and captions. Specifically, our results suggest text-only recoverability is a necessary (but not sufficient) condition for modeling compositional factors in contrastive VL models. We release our datasets and code.

## 1 Introduction

*"A penguin on Mars wearing a spacesuit and walking a robot dog next to Santa Claus."* Riedl (2022)'s text-to-image query is the type that modern multimodal models *should* be able to support. It is spatially precise (the dog is *next to* Santa, not in front), compositional (robot dog, but not robot Santa), and imaginative (it is unlikely such an image exists already). However, several recent works have shown that a variety of multimodal models (despite achieving strong benchmark performance) are frequently unable to reason about even simple spatial relations or attribute attachments (Gokhale et al., 2022; Thrush et al., 2022; Yuksekgonul et al., 2023).

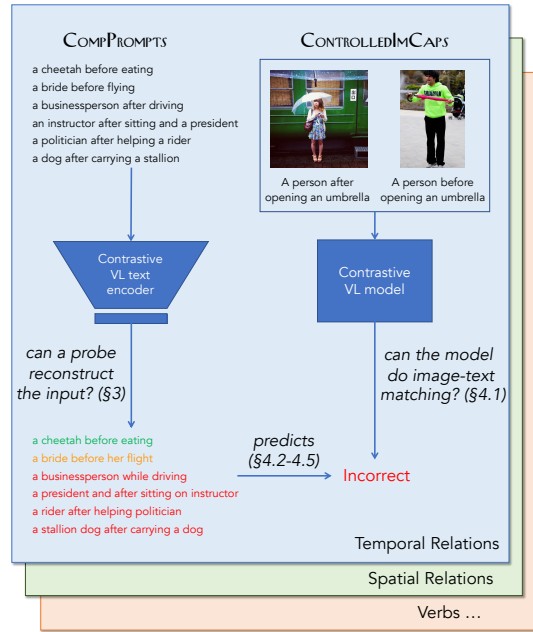

Figure 1: We present COMPPROMPTS, a dataset of 18,100 text prompts, and CONTROLLEDIMCAPS, a dataset of 600 image pairs+captions that differ by only one word. The two datasets are grouped by the same set of caption properties, e.g., temporal/spatial relations. Experiments on COMPPROMPTS quantify the information loss of a text encoder; experiments on CONTROLLEDIMCAPS illustrate that information loss correlates with multimodal errors.

Underlying several popular multimodal models like CLIP (Radford et al., 2021), DALL-E 2 (Ramesh et al., 2022) and ALIGN (Jia et al., 2021) is a *pooled text encoder,* i.e., a text representation model that outputs a single vector for a given input caption.[1] In this work, we use this representational bottleneck as an interface to ask: how precise are textual representations of visually-descriptive language in these modern multimodal models?

---

[1] Pooled text encoders (c.f., bidirectional multimodal encoders) are used for a variety of practical reasons: e.g., for guided diffusion (Dhariwal and Nichol, 2021), for fast k-NN queries over billions of images (Schuhmann et al., 2022), for contrastive objectives dependent on large batch sizes like Radford et al. (2021)'s 32K example "mini"-batch, etc.

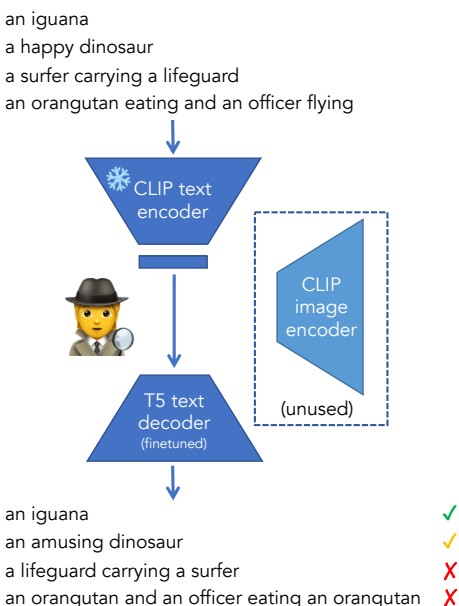

an iguana
a happy dinosaur
a surfer carrying a lifeguard
an orangutan eating and an officer flying

CLIP text encoder

CLIP image encoder

(unused)

T5 text decoder
(finetuned)

an iguana ✓
an amusing dinosaur ✓
a lifeguard carrying a surfer ✗
an orangutan and an officer eating an orangutan ✗

Figure 2: We probe the representations of single-vector text encoders used in popular VL models. Using a corpus of increasingly compositional image captions, COMPPROMPTS, we attempt to generatively decode the original input sentence. Text encoders of popular models like CLIP fail to effectively encode precise aspects of their captions like attribute attachments and object relationships (real examples shown, as in Figure 1).

Our strategy is as follows: for a fixed pooled text encoder $\mathcal{T} : x \rightarrow y$, which maps from captions $x$ to vectors $y \in \mathbb{R}^d$, we test how accurately $x$ can be recovered by an expressive generative decoder given $y$, i.e., $\mathcal{P}(x|\mathcal{T}(x))$. In an ideal case, $\mathcal{T}$ should result in *no* information loss, i.e., an exact reconstruction of the original caption should be possible, to account for specific visual factors. However, we hypothesize that if a specific visually descriptive property (e.g., a spatial relation) cannot be accurately decoded from $y$ (using a decoder trained with significant supervision), then it is unlikely a multimodal model can effectively use that property of $x$ using $\mathcal{T}$. Different from existing probes, ours does not require images, enabling exploration of a broader range of captions, e.g., creative text-to-image queries for which there may be no associated image (like *"A penguin on Mars..."*).

We execute our probe using an increasingly-compositional hierarchy of image captions we curate, COMPPROMPTS, which covers cases ranging from a single object with no attributes (e.g. "a cat") to multiple objects with attributes and relations (e.g. "an orange cat to the left of a dog"). We also test counting (e.g. "three cats") (Seguí et al.,

2015; Parcalabescu et al., 2021) and negations (e.g. "a cat that is not yawning"). We compare five text encoders, and find that top contrastive VL models: (1) are broadly ineffective at textually encoding spatial relations, numbers, and negations; (2) frequently cannot match attributes to their corresponding objects; and (3) fail more as inputs grow more compositional. While some text encoders perform significantly better than others, all underperform a proof-of-concept model which demonstrates that our prompts can indeed be compressed into single vectors with little information loss.

In order to verify that our text-only probe predicts performance in multimodal settings, we curate an evaluation set of image-caption pairs, CON~ TROLLEDIMCAPS, which operationalizes the compositional factors of COMPPROMPTS in a multimodal setting. Results on this corpus suggest our text-only probe gives a *necessary* condition: if the text-only recovery probe fails to recover a text-only property on COMPPROMPTS, then the associated multimodal model also performs poorly for that property on CONTROLLEDIMCAPS. However, our results also suggest that text-only recoverability is not a *sufficient* condition: a model can achieve low text-only information loss on a particular prompt type but not fully solve it on CON~ TROLLEDIMCAPS. To facilitate future probing experiments, we release our code alongside the newly collected COMPPROMPTS and CONTROLLEDIMCAPS corpora at https://github.com/amitakamath/vl_text_encoders_are_bottlenecks.

## 2 Evaluation Corpora

### 2.1 COMPPROMPTS

We create an evaluation dataset of 18,100 text prompts describing potential visual scenes with varying degrees of specificity and composition. Our starting point is animate nouns with corresponding verbs and adjectives from the Web10K dataset (Kamath et al., 2022). We remove some synonyms to prevent ambiguity in the prompt (e.g. "a rhino to the left of a rhinoceros").

The prompts are increasingly compositional: They have 1 or 2 unique nouns, and 0, 1, or 2 attributes, of which there are 4 types: adjective, verb, spatial, and temporal. Nouns are randomly matched to generate prompts with two unique nouns — this results in unusual and imaginative text inputs that cannot be guessed based on priors learned during model pre-training (e.g., "a crab lift-

ing a rhino"). The verb and spatial attributes can have either one associated noun (i.e. intransitive, e.g. "a koala yawning", "a policeman on the left") or two (i.e. transitive, e.g. "a poet chasing a rabbit", "a dinosaur left of a tiger"). We also test multiples and negations in the one-attribute setting.

Prompt examples of each type are given in Tables 2 and 3. There are 300-500 examples of each prompt type in the dataset.

## 2.2 CONTROLLEDIMCAPS

We create a second evaluation dataset to evaluate the overall vision-language model, rather than the text encoder specifically: where COMPPROMPTS contains text prompts alone, CONTROLLEDIMCAPS contains 600 pairs of images, along with a corresponding caption for each image.

The images are sourced from the COCO validation set (Lin et al., 2014), and the captions are hand-written to study one of six specific fine-grained textual properties: spatial relations with one associated noun, spatial relations with two associated nouns, temporal relations, verbs with one associated noun, verbs with two associated nouns, or adjectives. For spatial relations, we evaluate only "left" and "right" (unlike COMPPROMPTS, which evaluates also "above", "under", "in front of", and "behind"), due to insufficient presence of other spatial relations clearly depicted in the COCO data.

A key property of CONTROLLEDIMCAPS is that *only one word changes* between the two captions associated with a given image pair, such that the relation is changed or inverted: e.g., the caption pair "a person before opening an umbrella", "a person after opening an umbrella", along with the corresponding images for each (as in Figure 1) tests the overall model's understanding of temporal relations alone, without conflating any other types of reasoning.

## 3 Text-only Recovery

For a given text encoder $\mathcal{T}$, our first step is to obtain a training corpus of representations to fit a decoding probe $\mathcal{P}(x|\mathcal{T}(x))$. We use (just the text of) Conceptual Captions 3M (Sharma et al., 2018) (CC3M) split into a 90/10 train/val set; this corpus consists of cleaned alt-texts from web images, and thus is similar to the pretraining corpora of many VL models. For $\mathcal{P}$, we use T5-large: specifically, we condition the decoder on $\mathcal{T}(x)$, followed by a linear transformation and layer normalization. We train using Adafactor (Shazeer and Stern, 2018) with

| $\mathcal{T}(x)$ | Embed. size | Avg. EM (%) |
|---|---|---|
| CLIP ViT-B/32 | 512 | 13.2 |
| CLIP ViT-L/14 | 768 | 28.5 |
| negCLIP ViT-B/32 | 512 | 28.6 |
| RoBERTaCLIP ViT-B/32 | 512 | 28.9 |
| SBERT | 768 | **41.6** |
| Proof-of-concepT5 | 1024 | 92.9 |

Table 1: Average EM performance of each text encoder on COMPPROMPTS, not including multiples and negations (reported in Table 3).

a batch size of 512 for 4 epochs over CC3M; we select checkpoints with the lowest val loss. Models are trained using 4xA6000 GPUs with 48GB of memory using Transformers (Wolf et al., 2019) and accelerate[2]. At evaluation time, we generate captions for COMPPROMPTS set using beam=5 search.

**Text Models.** We evaluate several $\mathcal{T}$ models: CLIP ViT-B/32 (12 layers, 512 dim) and ViT-L/14 (12 layers, 768 dim) (Radford et al., 2021), CLIP with a RoBERTa-pretrained text encoder (Liu et al., 2019; Ilharco et al., 2021), and Yuksekgonul et al. (2023)'s more-order-aware CLIP encoder finetuned with hard negatives, negCLIP. For comparison, we also consider the uni-modal SentenceBERT (Reimers and Gurevych, 2019) model all-mpnet-base-v2, which is trained on several sentence similarity datasets including COCO captions (Lin et al., 2014).

**Proof-of-concepT5** We also consider a T5-large text encoder that produces a single vector output via mean pooling over the token embeddings. In contrast to the other fixed encoders, we fine-tune this model on CC3M, like an autoencoder[3]. Then, we use the resulting encoder as a feature extractor, and hand a dimension-shuffled version of the resulting embeddings to the probe. This "proof of concept" encoder is specifically optimized to generate a vector from which a T5 model can decode the full sentence, and serves to validate that our probe setup is even possible.

**Evaluation.** We evaluate using exact match (EM). While we report BLEU scores in the Appendix, for our high-precision setting, partial credit metrics are too generous, e.g., generating "a re-

---

[2]https://github.com/huggingface/accelerate
[3]There is no overlap between CC3M and COMPPROMPTS.

| | 0 attributes | 1 attribute | | | | 2 attributes | | | | | | | | | |
|---|---|---|---|---|---|---|---|---|---|---|---|---|---|---|---|
| | | 1 adj. | 1 spatial | 1 1-obj verb | 1 2-obj verb | 2 adj. | 1 adj + 1 spatial | 1 adj + 1 1-obj verb | 1 adj + 1 2-obj verb | 1 spatial + 1 1-obj verb | 1 spatial + 1 2-obj verb | 2 spatial | 1 temp. + 1 1-obj verb | 1 temp. + 1 2-obj verb | 2 verbs |
| **1 unique object** | *a cat* | *an orange cat* | *a cat on the left* | *a cat yawning* | - | *an orange and spotted cat* | *an orange cat on the left* | *an orange cat yawning* | - | *a cat on the left yawning* | - | - | *a cat before yawning* | - | - |
| CLIP ViT-B/32 | 3.0 | 38.2 | 34.6 | 15.4 | | 14.4 | 47.0 | 36.4 | | 15.6 | | | 17.0 | | |
| CLIP ViT-L/14 | 33.0 | 81.8 | 53.2 | 71.6 | | 23.2 | 43.8 | 62.6 | | 6.2 | | | 58.6 | | |
| negCLIP ViT-B/32 | 2.3 | 42.2 | 57.2 | 20.6 | | 20.8 | 63.6 | 50.4 | | 22.8 | | | 42.4 | | |
| RoBERTa-CLIP ViT-B/32 | 1.3 | 17.0 | 89.4 | 30.0 | | 42.2 | 83.4 | 59.8 | | 5.6 | | | 28.4 | | |
| SBERT | 54.0 | 91.8 | 91.8 | 78.4 | | 35.6 | 85.6 | 76.8 | | 21.2 | | | 57.6 | | |
| **2 unique objects** | *a cat and a dog* | *an orange cat and a dog / a cat and a brown dog* | *a cat to the left of a dog* | *a cat yawning and a dog* | *a cat chasing a dog* | *an orange cat and a brown dog* | *an orange cat to the left of a dog* | *a cat yawning and a brown dog* | *a cat chasing a brown dog / an orange cat chasing a dog* | *a cat yawning to the left of a dog* | *a cat chasing a dog on the left* | *a cat on the right and a dog on the left* | *a cat before yawning and a dog* | *a cat before chasing a dog* | *a cat yawning and a dog stretching* |
| CLIP ViT-B/32 | 13.4 | 15.8 | 6.2 | 1.6 | 6.6 | 17.8 | 6.2 | 0.8 | 7.2 | 8.8 | 4.2 | 1.2 | 1.6 | 3.4 | 1.8 |
| CLIP ViT-L/14 | 48.4 | 30.2 | 17.8 | 5.0 | 37.2 | 23.6 | 14.0 | 1.4 | 21.0 | 16.6 | 7.8 | 0.6 | 0.2 | 21.6 | 4.8 |
| negCLIP ViT-B/32 | 52.8 | 41.4 | 36.2 | 12.2 | 39.4 | 35.6 | 24.8 | 4.8 | 24.2 | 28.6 | 16.4 | 12.4 | 7.4 | 19.2 | 8.8 |
| RoBERTa-CLIP ViT-B/32 | 47.6 | 29.0 | 34.0 | 14.6 | 48.8 | 24.0 | 20.7 | 8.0 | 29.8 | 30.5 | 39.2 | 5.6 | 0.4 | 26.8 | 1.6 |
| SBERT | 56.0 | 41.8 | 44.8 | 11.2 | 48.8 | 32.8 | 30.3 | 8.0 | 30.2 | 31.9 | 23.8 | 3.0 | 4.4 | 4.8 | 10.8 |

Table 2: Prompt example and exact match (% EM) score of reconstruction from all models, averaged over several hundred instances each. As inputs become more compositional, vision-language text encoders perform increasingly poorly on text reconstruction.

porter on top of a penguin" as "a penguin on top of a hill" scores 48 BLEU-4 points. Similarly for BERT-Score (Zhang et al., 2020), where generating "two rabbits and three shrimps" as "four of the shrimps and a rabbit" scores 0.91 F1.

## 3.1 Text-only Recovery Results

Table 1 presents the average exact match of each model over the corpus of prompts in CompPrompts, excluding negations and multiples, which are reported in Table 3. The proof-of-concepT5 model's high performance illustrates that it is possible in theory to nearly exactly decode all captions in CompPrompts using T5-large, given the "right" encoding[4]. Beyond proof-of-concepT5, the best performing model is SBERT; and the best performing multimodal model is RoBERTa-CLIP.

## 3.2 Fine-Grained Results on Different Prompt Types

Table 2 contains EM results of all models on the various types of prompts in CompPrompts. A separate study on multiples and negations in Table 3 shows that text encoders struggle to encode those as well. These results show that it is fairly difficult to decode input sentences from text representations for most VL models, with increasingly compositional categories proving more difficult (e.g., "an orange cat" to "an orange cat yawning" to "an orange cat chasing a dog").

**Spatial relations.** Text encoders of VL models struggle to represent spatial relations (average 23.7

EM), particularly those between two objects (average 13.8 EM). SBERT, in comparison, scores 36.9 and 22.3 EM, respectively.

**Temporal relations.** VL models perform poorly on temporal relations, scoring on average 17.1 EM. In comparison, SBERT scores 29.6 EM — likely because temporal relations appear more frequently in language than in web alt-text.

**Transitive vs intransitive verbs and prepositions.** On transitive verbs (e.g., "chasing"), CLIP ViT-B/32 and ViT-L/14 do worse by an average of 21 EM than vs. intransitive verbs (e.g., "yawning"), whereas negCLIP and RoBERTa-CLIP do better by an averaged 18.7 points. On transitive prepositions ("to the left of") instead of intransitive ("on the left"), all models do worse by an averaged 35 EM.

**Negations and multiples.** Models perform poorly on negations (average EM 13.0) and multiples (average EM 5.1). This agrees with previous observations that VL models struggle with counting (Seguí et al., 2015; Parcalabescu et al., 2021).

**Prompts where word order matters.** VL text encoders struggle to capture word order: on prompts where word order matters less (e.g., "a cat and a dog"), they score an average of 34 EM, but where word order matters more, they score an average of 15.8 EM. The failure cases are often caused by assigning attributes to nouns incorrectly, as highlighted in the Appendix. This extends Thrush et al. (2022)'s and Yuksekgonul et al. (2023)'s finding that contrastive VL models can behave like bags-of-words — this issue manifests just in the text encoder as well.

---

[4]Most errors made by proof-of-concepT5 are minor e.g., "two physicians on the right" → "two physician on the right".

| | 0 attributes | 1 attribute | | | |
|---|---|---|---|---|---|
| | | 1 adj. | 1 spatial | 1 1-obj verb | 1 2-obj verb |
| **1 unique object + multiples** | *two cats* | *two orange cats* | *two cats on the left* | *two cats yawning* | - |
| CLIP ViT-B/32 | 0.3 | 15.0 | 9.6 | 9.4 | |
| CLIP ViT-L/14 | 3.0 | 14.8 | 10.8 | 16.8 | |
| negCLIP ViT-B/32 | 1.0 | 9.4 | 11.2 | 12.8 | |
| RoBERTa CLIP ViT-B/32 | 0.7 | 4.6 | 16.0 | 6.4 | |
| SBERT | 37.3 | 56.6 | 48.4 | 33.2 | |
| **2 unique objects + multiples** | *two cats and four dogs* | - | *two cats to the left of four dogs* | - | *two cats chasing four dogs* |
| CLIP ViT-B/32 | 0.0 | | 0.0 | | 0.0 |
| CLIP ViT-L/14 | 0.0 | | 0.0 | | 0.0 |
| negCLIP ViT-B/32 | 0.0 | | 0.0 | | 0.2 |
| RoBERTa CLIP ViT-B/32 | 0.6 | | 0.2 | | 0.6 |
| SBERT | 0.0 | | 0.0 | | 0.0 |
| **1 unique object + negation** | - | *a cat that is not orange* | *a cat that is not on the left* | *a cat that is not yawning* | - |
| CLIP ViT-B/32 | | 18.0 | 7.4 | 6.8 | |
| CLIP ViT-L/14 | | 10.4 | 6.4 | 7.8 | |
| negCLIP ViT-B/32 | | 1.8 | 2.2 | 14.2 | |
| RoBERTa CLIP ViT-B/32 | | 43.8 | 50.4 | 58.4 | |
| SBERT | | 32.8 | 4.6 | 19.4 | |
| **2 unique objects + negation** | - | - | *a cat that is not to the left of a dog* | - | *a cat that is not chasing a dog* |
| CLIP ViT-B/32 | | | 0.6 | | 0.8 |
| CLIP ViT-L/14 | | | 0.4 | | 0.4 |
| negCLIP ViT-B/32 | | | 2.6 | | 1.8 |
| RoBERTa CLIP ViT-B/32 | | | 12.6 | | 13.6 |
| SBERT | | | 2.6 | | 2.6 |

Table 3: All models' EM on prompts that contain multiples or negations. Text recovery of these inputs is very poor, likely because multiples and negations tend to be infrequent in image captions.

**Adjectives and verbs.** VL models perform relatively well in the basic one-object, one-attribute setting on both adjectives (average EM 44.8) and verbs (average EM 34.5): even higher than the zero-attribute setting, where error analysis reveals they tend to hallucinate information ("a tarantula" → "a tarantula in a hand"). While these numbers are well behind SBERT (EM 91.8 and 78.4 respectively), they agree with previous observations that VL models exhibit good visual recognition of basic adjectives and actions (Radford et al., 2021).

**Compositionality.** Text encoders struggle with increasingly compositional information, e.g., the probe decodes SBERT("a dentist after examining an ape") → "an ape after examining a dentist". On average, performance on two unique objects drops by 49% from their performance on one unique object (for CLIP ViT-B/32, it drops 71%). VL model performance drops on average by 35% when the prompt contains two attributes compared to one.

### 3.3 Fine-Grained Results for Different Model Designs

**Pre-training the text encoder helps, especially on negations.** The average EM of RoBERTa-CLIP on prompts without multiples or negations is 15.7 points higher than CLIP ViT-B/32. However, on the prompts that do include negations, its average EM is 29 points higher. This provides evidence that text pre-training the text encoder helps negations, presumably because negations are less likely in alt-texts compared to other settings.

**Increasing model size helps overall, but not on spatial relations.** The average EM of CLIP ViT-L/14 on prompts that do not include spatial relations is 20.7 points higher than CLIP ViT-B/32. However, on the prompts that do include spatial relations, its average EM is only 4 points higher. The modest increase of text encoder size in the CLIP training regime appear less reliable for encoding spatial relations than text pre-training or hard negatives (though, more significant scaling could be beneficial, as in Imagen (Saharia et al., 2022)).

**Hard negatives from Yuksekgonul et al. (2023) help, especially where word order matters.** On average, negCLIP does 15.4 points better than CLIP. On prompts where word order matters (e.g. "a cat chasing a dog"), it scores 16.3 points higher; on prompts where word order does not matter (e.g. "a cat and a dog"), it scores 12.8 points higher.

### 3.4 Incorrect Model Predictions

We manually inspect models' incorrect predictions. Decoded VL text encoder predictions often come close (e.g. "three shrimps" → "three of shrimp" is a pattern shown by CLIP ViT-B/32, CLIP ViT-L/14 and negCLIP), whereas SBERT's incorrect decodings fall further afield (e.g. "three gardeners" → "three gardeners and a third man."). Thus, while the superior results of the unimodal SBERT compared to the VL text encoders when evaluated in the same frozen-encoder setting (including CLIP ViT-L/14, which has the same text embedding size) show that there is significant room for improvement for VL text encoders, the types of errors made by each model may not be fully captured by EM. Nonetheless, EM remains an appropriate metric for our high-precision setting, as discussed in Section 3.

## 4 Experiments and Results in the Multi-modal Setting

We investigate the hypothesis that if a textual property cannot be decoded from the VL text encoder's vector representation with a highly expressive decoder (like T5-Large), then it also cannot be readily modeled in the multimodal setting. Controlled~ ImCaps studies the attributes from CompPrompts in the multimodal setting. We then compare the text

| Type | Comparable Prompts in CompPrompts | Example from ControlledImCaps |
|---|---|---|
| Spatial-1 | 1 unique object + 1 1-obj preposition (Left/Right only) | a cat on the right / a cat on the left |
| Spatial-2 | 2 unique objects + 1 2-obj preposition (Left/Right only) | a person to the right of a horse / a person to the left of a horse |
| Temporal | 1 unique object + temporal relation of a 1-obj verb, 2 unique objects + temporal relation of a 1-obj verb, 2 unique object + temporal relation of a 2-obj verb | a dog before catching a frisbee / a dog after catching a frisbee |
| Verb-1 | 1 unique object + 1 1-obj verb | a bird sitting / a bird flying |
| Verb-2 | 2 unique objects + 1 2-obj verb | a person feeding an elephant / a person riding an elephant |
| Adjective | 1 unique object + 1 adjective | a blue fire hydrant / a white fire hydrant |

Figure 3: Each attribute in ControlledImCaps, with comparable prompts in CompPrompts and an example.

| Prompt Type | EM on CompPrompts | CIC Image score | CIC Text score |
|---|---|---|---|
| Spatial 1-obj L/R | 28.5 | 4.0 | 15.0 |
| Spatial 2-obj L/R | 4.4 | 4.0 | 8.0 |
| Temporal | 26.8 | 7.0 | 30.0 |
| Verb 1-obj | 71.6 | 84.0 | 89.0 |
| Verb 2-obj | 37.2 | 46.0 | 63.0 |
| Adjectives | 81.8 | 65.0 | 85.0 |

Table 4: Performance of CLIP ViT-L/14 text encoder (%) on the equivalent prompts in CompPrompts, and performance of CLIP ViT-L/14 full model on Controlled~ImCaps (CIC). On the types of prompts where the text encoder performs poorly, so too does the overall model.

than the incorrect caption when given an image, and an *image score*, the fraction of instances where a model scores the correct image higher than the incorrect image when given a caption.

### 4.1 Multi-modal Results

Table 4 presents the results of CLIP ViT-L/14 on both CompPrompts and ControlledImCaps (all model results in Appendix). The CompPrompts results correspond to the prompt type(s) most closely matching the captions in ControlledImCaps (specified in Figure 3). For the spatial relations, for this table alone, we calculate the EM on the data points in CompPrompts containing "left" and "right" spatial relations only due to lack of sufficient support in COCO for other spatial relations, as discussed in Section 2.2. On prompt types where the text encoder performance on CompPrompts is poor, the overall model performance on ControlledImCaps is also poor: showing that the text encoder does indeed bottleneck VL models' compositionality.

We see similar findings per prompt type and model design as those discussed in Section 3.3.

### 4.2 Fine-Grained Results on Different Prompt Types

We discuss findings on the prompt types in Con~trolledImCaps, with 95% confidence intervals.

**Models do poorly on spatial relations.** On average, VL models perform poorly on spatial relations, achieving an average image | text score of 2.5 | 12.4 (± 2.2 | 3.7). Their text encoder performance on the corresponding prompts in CompPrompts was similarly poor, with an average EM of 27.8. This agrees with Kamath et al. (2023), which shows that VL models struggle with spatial relations.

encoder performance of a VL model on a particular prompt type in CompPrompts with the performance of the overall VL model on that prompt type in ControlledImCaps. As discussed in Section 2.2, the two captions in every example differ by only one word which changes or inverts the relation, allowing us to perform fine-grained analyses in controlled settings without conflating multiple types of compositional reasoning. Figure 3 depicts the six types of attributes studied in ControlledImCaps, their corresponding prompt type in CompPrompts, and an example of each.

**VL Models.** We evaluate the same VL models as in Section 3: CLIP ViT-B/32, CLIP ViT-L/14, CLIP with a RoBERTa-pretrained text encoder (Liu et al., 2019; Ilharco et al., 2021), and negCLIP (Yuksekgonul et al., 2023). Each of these models can return a score when given an image and a caption, representing how well they match.

**Evaluation.** We follow the evaluation scheme from Winoground (Thrush et al., 2022): for a given pair of images with corresponding captions, we measure both a *text score*, the fraction of instances where a model scores the correct caption higher

**Models do poorly on temporal relations.** VL performs poorly on temporal relations, with an average image | text score of 5.3 | 30.8 ($\pm$ 2.7 | 4.8). Their text encoder performance on CompPrompts temporal reasoning was similarly low at 18.9 EM.

**Models do well on verbs and adjectives.** VL models perform well on verbs (average image | text score 65.4 | 78.1, $\pm$ 5.0 | 4.8) and even better on adjectives (average image | text score 78.5 | 89.0, $\pm$ 7.0 | 3.5), mirroring their text encoder performance on CompPrompts, where the average EM for verbs and adjectives were 33.7 and 44.8 respectively.

**Two-object verbs are more difficult than one-object verbs.** We find that for all models, two-object verbs are harder than one-object verbs, with the former achieving an image | text score of 52.3 | 68.5 and the latter 78.5 | 87.8 (with $p < 0.05$ under the Wilcoxon signed-rank test). This follows performance on CompPrompts for ViT-B/32 and ViT-L/14, but not for negCLIP and RoBERTa-CLIP, hinting that ability to reconstruct is necessary but not sufficient, as discussed in Section 4.5.

### 4.3 Fine-grained results on different model design choices

We discuss findings on the model designs in ControlledImCaps. All findings are statistically significant at $p < 0.05$ using the Wilcoxon signed-rank test to compare models.

**Pre-training the text encoder improves text score on verbs.** RoBERTa-CLIP obtains a higher text score than CLIP ViT-B/32 (78.0 vs 68.0), as well as a higher EM on the prompts in CompPrompts corresponding to verbs (39.4 vs 11.0).

**Increasing model size does not help on spatial or temporal reasoning.** On both spatial and temporal reasoning inputs, ViT-L/14 performance on ControlledImCaps was not statistically significantly higher than that of ViT-B/32.

**Hard negatives from Yuksekgonul et al. (2023) help where word order matters.** On prompts where word order matters, negCLIP scores an image | text score of 36.5 | 50.0 and a CompPrompts EM of 27.2, and other models score an average image | text score of 24.8 | 37.5 and a CompPrompts EM of 21.0. negCLIP also outperforms ViT-B/32 on all prompts on average.

### 4.4 Text reconstruction appears to be necessary...

To study the relationship between text reconstruction and overall model performance beyond Table 4, we evaluate text reconstruction on ControlledImCaps. Specifically, we use the trained T5 decoders from Section 3 and try to reconstruct the input when ControlledImCaps text inputs are evaluated. On the cases where the reconstruction is *incorrect* according to human evaluation[5] on either of the two text inputs, the overall model Image Score on ControlledImCaps for CLIP ViT-L/14 is zero 96% of the time, and the Text Score is zero 83% of the time. This text reconstruction vs. multimodal matching correlation is more direct compared to the similar correlation reported in Table 4 because we compare on the same instances.

### 4.5 ... but insufficient.

Conversely, just because a model performs well on the CompPrompts probe does not mean it will perform well on ControlledImCaps. For example, ViT-L/14 outperformed ViT-B/32 overall on CompPrompts, but not (statistically significantly) on ControlledImCaps. Also, RoBERTa-CLIP outperforms ViT-B/32 on temporal relations on CompPrompts, but achieves a worse text score on ControlledImCaps. When we evaluate text reconstruction on ControlledImCaps, on cases where the reconstruction is *correct* for both text inputs, the overall model Image Score on ControlledImCaps for ViT-L/14 is zero 59% of the time, and the Text Score is zero 47% of the time. This suggests that text recoverability is a necessary but insufficient condition for overall model performance. The insufficiency is intuitive, as multimodal errors could potentially stem from the image encoder.

### 4.6 A Note on Winoground

We evaluate our four VL models on the Winoground dataset (Thrush et al., 2022). They perform poorly, with an average image | text score of 10.3 | 30.8, where random chance is 25.0 | 25.0. However, on shorter inputs (5 words or less) which exhibit fewer compositional concepts on average, e.g., "a bird eats a snake" | "a snake eats a bird", the four models achieve higher scores of 20.4 | 47.2 on average. On longer (over 10 words), more

---

[5]For the simple inputs of ControlledImCaps, we found human evaluation by the authors tractable, with the added advantage of not penalizing minor errors as EM does.

compositional inputs, e.g. "in the stadium, the person wearing gray outperformed the one wearing blue" | "in the stadium, the person wearing blue outperformed the one wearing gray", models achieve a much lower score of 3.4 | 18.5. This mirrors our finding on CompPrompts that VL text encoders struggle with increasingly compositional inputs.

## 5    Related work

Building models capable of reasoning jointly about visual and textual inputs is a long-standing goal of AI (Winograd, 1971), with potential applications in the fields of vision-language navigation (Anderson et al., 2018), human-robot interaction (Matuszek et al., 2012), accessible image captioning (Gurari et al., 2020), etc.

Recent challenge datasets have been designed to probe the capacity of multimodal models to represent descriptions of precise visual compositions (Johnson et al., 2017; Suhr et al., 2019; Hudson and Manning, 2019; Thrush et al., 2022). Yuksekgonul et al. (2023) and Yamada et al. (2022) study CLIP specifically, demonstrating its shortcomings (and some potential fixes) in terms of modeling syntax. Ma et al. (2022) study OpenCLIP models for various types of compositional reasoning, with programmatically sourced hard negatives. Different from these works, our textual probe does not require access to images.

Our image-and-text evaluation most closely resembles Thrush et al. (2022). However, we stratify the examples based on type of input (e.g., temporal relations) to provide more detailed insights. We also keep our prompts relatively simple, never having more than two objects or two attributes in the input. We believe this is a more realistic goal for our current vision-language models. The word order shuffling aspect is also discussed in Yuksekgonul et al. (2023). However, as their proposed benchmark does not provide pairs of images with corresponding captions, it is possible to achieve state-of-the-art with a text-only model (specifically, 2-shot ChatGPT[6] (Ouyang et al., 2022), details in Appendix and the recent Hsieh et al. (2023)). While this does not detract from their finding that vision-language models ignore word order, our benchmarks have an additional advantage of being insensitive to text-only priors.

---
[6]https://platform.openai.com/docs/api-reference/chat, using the gpt-3.5-turbo model

## 6    Conclusion and Discussion

We present probing results that suggest significant information loss upon text encoding of compositional inputs in vision and language models. This information loss is quantified using CompPrompts, a test set of increasingly compositional image descriptions, and ControlledImCaps, a test set that we use to verify that this information loss affects the performance of multimodal models on compositional inputs. Harder negatives, more text pretraining, and larger models all improve encoder quality, but information is still lost even for the most performant models, compared to the unimodal SBERT as well as a T5-based auto-encoder.

Going forward, even more difficult test sets than CompPrompts and ControlledImCaps might be required to analyze and evaluate vision-language model capabilities. Returning to Riedl (2022)'s tweet from the intro, *"A penguin on Mars wearing a spacesuit and walking a robot dog next to Santa Claus."*, even our highly accurate proof-of-concepT5 model struggles, predicting: *"compulsory penguin onexposition wearing a spacesuit and walking a dog robot next tohoc"*. To support imaginative text-to-image generation queries (for images that may not exist yet), future work would be well-suited to design text encoders that can generalize to captions that contain compositions never-before-seen in web alt-text corpora.

Our probing results suggest two future modeling directions: (1) Modifying contrastive VL models' training objectives to additionally encourage ability to reconstruct the text input, either through an additional reconstruction loss on the text encoder during finetuning, or through the addition of even harder negatives than Yuksekgonul et al. (2023) and Ma et al. (2022), would be an exciting avenue for future work. Alternatives to contrastive training, such as captioning, have also shown promise in recent work (Tschannen et al., 2023); and (2) explicitly encouraging linear recovery with a modified loss function: while the gap between SBERT and the VL Text encoders can be partially explained by the superior pooling method and training data, SBERT's training objective does not require linear recoverability (whereas CLIP's dot product interaction term might): explicitly encouraging linear text-text recoverability might improve multimodal performance. Finally, we hope that ControlledIm~Caps can facilitate research beyond single-vector bottleneck VL models.

## Limitations

First, our probing method involves a pre-trained T5 decoder. It is possible that language biases from the pre-training emerge while decoding from the VL text embedding, e.g., predicting "a dog chasing a cat" instead of "a cat chasing a dog" because the former is more likely under the T5 decoder's priors from pre-training. However, as the methodology is the same across all models we evaluate, we believe that the evaluation is fair. Second, we evaluate with only one probe, whereas probing with complementary methods (e.g., especially deterministic ones, like a convex linear probe) could reveal more insights. Third, text encoders that do well on our evaluation may not perform well if directly plugged into a contrastive VL model like CLIP, if the text encoders were not trained to encode the information in a manner that is linearly recoverable.

## Acknowledgements

The authors thank John Hewitt, Akhila Yerukola, and anonymous reviewers for useful discussion and feedback. This work was funded by the Allen Institute for AI. AK was additionally supported by the UCLA Computer Science Department First-Year Fellowship. KC was supported in part by U.S. DARPA MCS Program under contract number N660011924032, U.S. DARPA ECOLE Program No. HR00112390060, and ONR N00014-23-1-2780, and a Sloan Fellowship. The views and conclusions contained herein are those of the authors and should not be interpreted as necessarily representing the official policies, either expressed or implied, of DARPA, or the U.S. Government.

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

## A  Additional Results

Table 5 contains average BLEU-4 scores of the models. Table 6 contains a study of model performance on object-attribute association in COMP~PROMPTS. Table 7, Table 8 and Table 9 contain results of other models on CONTROLLEDIMCAPS in comparison to COMPPROMPTS (ViT-L/14 is discussed in Table 4). Table 10 discusses text-only results on the ARO benchmark (Yuksekgonul et al., 2023).

| $\mathcal{T}(x)$ | Embed. size | Avg. BLEU-4 |
|---|---|---|
| CLIP ViT-B/32 | 512 | 33.3 |
| CLIP ViT-L/14 | 768 | 46.0 |
| negCLIP ViT-B/32 | 512 | 50.2 |
| RoBERTaCLIP ViT-B/32 | 512 | 52.0 |
| SBERT | 768 | **56.7** |

Table 5: Average BLEU-4 performance of each text encoder on COMPPROMPTS, not including multiples and negations. The trend correlates with EM %, but the evaluation itself is too lenient for our purposes, as described in the main text.

| $\mathcal{T}(x)$ | Embed. size | Shuffled % ($\downarrow$) |
|---|---|---|
| CLIP ViT-B/32 | 512 | 51.8 |
| CLIP ViT-L/14 | 768 | 55.5 |
| negCLIP ViT-B/32 | 512 | **37.2** |
| RoBERTaCLIP ViT-B/32 | 512 | 62.8 |
| SBERT | 768 | 44.2 |

Table 6: Of the times the model gets the words in the prediction correct, Shuffled % is the percentage of when it gets the word order incorrect (in the prompts where word order matters, unlike "cat and dog" — specifically, where attributes must be associated with the correct object). Clearly, negCLIP having been trained with hard negatives involving word order shuffling allows it to perform the best. All models suffer from poor object attribute association.

| Prompt Type | EM on COMPPROMPTS | CIC Image score | CIC Text score |
|---|---|---|---|
| Spatial 1-obj L/R | 27.2 | 1.0 | 10.0 |
| Spatial 2-obj L/R | 0.6 | 4.0 | 10.0 |
| Temporal | 7.3 | 4.0 | 35.0 |
| Verb 1-obj | 15.4 | 71.0 | 77.0 |
| Verb 2-obj | 6.6 | 43.0 | 59.0 |
| Adjectives | 38.2 | 74.0 | 92.0 |

Table 7: Performance of CLIP ViT-B/32 text encoder (%) on the equivalent prompts in COMPPROMPTS, and performance of CLIP ViT-B/32 full model on CONTROLLED~IMCAPS (CIC).

| Prompt Type | EM on COMPPROMPTS | CIC Image score | CIC Text score |
|---|---|---|---|
| Spatial 1-obj L/R | 26.0 | 1.0 | 13.0 |
| Spatial 2-obj L/R | 15.0 | 3.0 | 13.0 |
| Temporal | 23.0 | 8.0 | 35.0 |
| Verb 1-obj | 20.6 | 84.0 | 94.0 |
| Verb 2-obj | 39.4 | 70.0 | 87.0 |
| Adjectives | 42.2 | 95.0 | 92.0 |

Table 8: Performance of negCLIP ViT-B/32 text encoder (%) on the equivalent prompts in COMPPROMPTS, and performance of negCLIP ViT-B/32 full model on CONTROLLEDIMCAPS (CIC).

| Prompt Type | EM on COMPPROMPTS | CIC Image score | CIC Text score |
|---|---|---|---|
| Spatial 1-obj L/R | 92.3 | 1.0 | 10.0 |
| Spatial 2-obj L/R | 28.3 | 2.0 | 20.0 |
| Temporal | 18.5 | 2.0 | 23.0 |
| Verb 1-obj | 30 | 75.0 | 91.0 |
| Verb 2-obj | 48.8 | 50.0 | 65.0 |
| Adjectives | 17 | 80.0 | 87.0 |

Table 9: Performance of RoBERTa-CLIP ViT-B/32 text encoder (%) on the equivalent prompts in COMP~PROMPTS, and performance of RoBERTa-CLIP ViT-B/32 full model on CONTROLLEDIMCAPS (CIC).

| Dataset | negCLIP | ChatGPT 2-shot |
| --- | --- | --- |
| VG-Relation | 0.81 | 0.90 |
| VG-Attribution | 0.71 | 0.80 |
| Flickr30k-PRC | 0.91 | 0.86 |
| COCO-PRC ViT-B/32 | 0.86 | 0.86 |

Table 10: Performance of ChatGPT 2-shot on the ARO benchmark (Yuksekgonul et al., 2023). While the dataset was designed to test VL models' sensitivity to word order shuffling (which is orthogonal to text-only performance on the same data), the textual priors that exist in ARO (e.g., "horse eating grass" is more likely than "grass eating horse") are less relevant to the probing experiments for CompPrompts (because the probe must reconstruct *any* given caption in CompPrompts, including unusual ones, e.g., "five teenagers riding three butterflies") and do not exist in ControlledImCaps due to the paired-image construction.