# OpenReview forum: "Text encoders bottleneck compositionality in contrastive vision-language models"
_EMNLP/2023/Conference — EMNLP 2023 Main_

### Official Review · Reviewer_n92c · 2023-08-02

**Soundness:** 4

**Excitement:**

4: Strong: This paper deepens the understanding of some phenomenon or lowers the barriers to an existing research direction.

**Paper Topic And Main Contributions:**

This paper reveals a significant language information loss in text encoders of contrastive language-image pretraining models (CLIP).

A text-decoding probing method is introduced, where the encoded CLIP text vector is fed into a generative LM for decoding the original captions, along with a prompt benchmark designed carefully to examine different aspects of objects, attributes & relations between objects. The evaluation results indicate that the text vectors of CLIP struggle to encapsulate spatial, and temporal relations, while do capturing simple attribute knowledge.

The paper further probes the performance under a multi-modal setting, similar to the setup of Winoground. It shows that the lower text representation indeed bottlenecks the image-text matching, echos most findings in the text-only setup and suggests that the ability to recover the text input is a necessary but insufficient condition for overall results.

**Questions For The Authors:**

- Did you explore other CLIP-variants, such as DeCLIP ( + language supervision),  ALBEF, and BLIP, on these probing tasks?  which may tell followers that some directions are useful while some are not.
- Regarding the objects,  the frequency in the corpus may introduce a significant effect on the performance. Maybe categorize the prompt according to the frequency and check the performance gap as well?

More discussions and results are always great but I not demanding these  :)  .

**Reasons To Accept:**

- The proposed probing method - text reconstruction - is interesting for probing the V+L performance.
- These findings are intuitively reasonable and insightful,  potentially inspiring future studies on improving the CLIP-like V+L models.
- The constructed benchmark released would be very useful.

**Reasons To Reject:**

- I do not see significant reasons to reject this paper.

**Reproducibility:**

4: Could mostly reproduce the results, but there may be some variation because of sample variance or minor variations in their interpretation of the protocol or method.

**Reviewer Confidence:**

4: Quite sure. I tried to check the important points carefully. It's unlikely, though conceivable, that I missed something that should affect my ratings.

**Typos Grammar Style And Presentation Improvements:**

- Table 3 can be turned in to a bar chart as there are so many blanks.

- Table 4: According to Line 375-379, the EM score of the first two rows is calculated differently from the rest rows. Maybe highlight this difference with * in the table?

---

> ### Author Rebuttal · Authors · 2023-08-29
>
> We thank all the reviewers for their detailed and constructive reviews. Thank you to Reviewer n92c for pointing out our interesting method, our reasonable and insightful findings potentially inspiring future work, and our useful benchmark.
>
> > **Exploring other CLIP-variants:**
> * For non-standard CLIP variants, we experimented with RoBERTaCLIP to determine the effect of text pre-training on the text encoder’s capabilities, and negCLIP to determine the effect of finetuning with hard negatives targeting word order shuffling. But as the reviewer correctly points out, exploring other variants of CLIP would be an interesting avenue for future work. Given our necessary recoverability condition, one could imagine future CLIP variants reporting text-only recoverability as a metric.
>
> > **Relative frequency of objects in the corpus:**
> * In CompPrompts, this is not an issue as the objects and attributes are selected at random from a set of such words. In expectation, there will be no major frequency variations between them.
> * In ControlledImCaps, this would not be an issue as the two captions in any data point vary only by one word, which is the phenomenon being tested (e.g. before/after for temporal prepositions).
>
> Thank you again for your thoughtful comments, and your suggestions to improve the readability of our results. Please let us know if you have any further questions or comments, we would be happy to follow up!

---

### Official Review · Reviewer_uEYG · 2023-08-03

**Soundness:** 4

**Excitement:**

4: Strong: This paper deepens the understanding of some phenomenon or lowers the barriers to an existing research direction.

**Paper Topic And Main Contributions:**

This paper presents a thorough examination of the information loss within the language module of CLIP, aiming to understand its impact on language processing. To facilitate their investigation, the researchers introduce two novel datasets, namely CompPrompts and ControlledImCaps. The compelling analysis leads them to a finding: the text encoders in CLIP act as performance bottlenecks, potentially affecting the model's overall language capabilities.

**Questions For The Authors:**

I am curious about the rationale behind utilizing the T5 decoder in the Text-only Recovery experiment. Wouldn't it be more appropriate to consider alternative decoder-only models, such as GPT-2, and evaluate their potential effectiveness in this context?

**Reasons To Accept:**

While the discovery that the text encoder in CLIP possesses limited language information may not be groundbreaking, this paper distinguishes itself through meticulous examination and the introduction of innovative datasets for their investigation. Moreover, the authors present an extensive array of experiment results to support their findings.

**Reasons To Reject:**

The revelation of the text encoder's limited capacity to retain language information in CLIP may not be considered groundbreaking.

**Reproducibility:**

5: Could easily reproduce the results.

**Reviewer Confidence:**

4: Quite sure. I tried to check the important points carefully. It's unlikely, though conceivable, that I missed something that should affect my ratings.

---

> ### Author Rebuttal · Authors · 2023-08-29
>
> We thank all the reviewers for their detailed and constructive reviews. Thank you to Reviewer uEYG for pointing out our thorough/meticulous examination, innovative datasets, and extensive experiments.
>
> > **The existence of a bottleneck is somewhat unsurprising.**
> * While it might not be surprising that text encoders bottleneck CLIP-style models to some extent, we are the first, to our knowledge, to quantify the (surprisingly significant!) information loss for various models. We are hopeful our work can inform future model design.
>
> > **Why T5 and not GPT-2:**
> * Decoder-only T5 and GPT-2 share very similar architectures: GPT-2 would also be a reasonable choice for our experiments. However, the high performance of proof-of-concepT5 shows that T5 is sufficient for the claims we make, i.e., that T5 is able to accurately recover captions if they are represented in the input features.
>
> Thank you again for your thoughtful comments. Please let us know if you have any further questions or comments, we would be happy to follow up!

---

### Official Review · Reviewer_iU94 · 2023-08-03

**Soundness:** 3

**Excitement:**

3: Ambivalent: It has merits (e.g., it reports state-of-the-art results, the idea is nice), but there are key weaknesses (e.g., it describes incremental work), and it can significantly benefit from another round of revision. However, I won't object to accepting it if my co-reviewers champion it.

**Paper Topic And Main Contributions:**

The paper addresses the problem of text encoders being a performance bottleneck in contrastive vision-language model.


The paper first proposes a text-only recovery method to reconstruct captions from single-vector text representations generated by
various VL models, and use it to evaluate the information loss caused by the text encoder on the CompPrompts dataset.
Additionally, it assesses how this information loss impacts the performance of VL models through the Controlled-ImCaps
benchmark.


The main contributions of this paper include:
1. A detailed analysis of the limitations of current text encoders in VL models.
2. The introduction of two new evaluation benchmarks, namely CompPrompts and Controlled-ImCaps, to respectively evaluate
information loss in text encoders and its effect on VL model performance.
3. The proposal of a text-only recovery method for reconstructing captions from single-vector text representations produced by
multiple VL models.
4. The identification that recoverability of textual content is necessary but not sufficient for modeling compositional factors in
contrastive VL models.


Furthermore, the paper provides an extensive review of related work and discusses potential future research directions in this
field.


**Reasons To Accept:**

The strengths of this paper include its comprehensive analysis of the limitations of current text encoders in contrastive VL models,
the proposal of two new evaluation benchmark called CompPrompts and Controlled-ImCaps, and the introduction of text-only
recovery method to evaluate the performance of text encoders in VL models on CompPrompts.

If this paper were to be presented at the conference or accepted into Findings, the main benefits to the NLP community would be
the insights it provides into the performance bottlenecks of text encoders in VL models and the proposed methods for evaluating
and improving their performance.

The paper's findings could inform the development of more effective text encoders for contrative VL models, which could have
applications in a wide range of areas. It may also aid the future work in designing text encoders that can generalize to captions that
contain compositions never-before-seen in web alt-text corpora and therefore, support imaginative text-to-image generation
queries (for images that may not exist yet).

Additionally, the proposed evaluation benchmarks could be used to compare the performance of different text encoders and VL models, which could facilitate the development of more accurate and robust models in this area.

**Reasons To Reject:**

The weakness of this paper is as follows:


1. In the experiments on the comp-prompts dataset, it fails to draw effective conclusions in a series of statements in the paper. For example, in Table 2, when comparing the EM scores for column 1 adj +1 spatial or column 1 adj +1-obj-verb with 2 attributes and 1 unique object to those for column 1 spatial or column 1 adj with only 1 attribute, it is found that CLIP ViT-B/32 has a higher score (47) in the former case than all three columns in the latter case. Similar observations can also be seen in negCLIP ViT-B/32.


2. Text recoverability of recovery-generated text is used to evaluate Text encoder's performance. Based on EM score metric, conclusions are drawn regarding Text encoder's shortcomings in different factors. However, there exists a gap between text encoder and T5 decoder since authors use decoder of Fine-tuned T5 model to reconstruct input text. I hope authors can provide an datailed explanation on how T5 decoder reconstructs input text from different text encoder embeddings especially when these embeddings have different sizes. Moreover, this gap may potentially affect the content of recovered texts. Since EM score serves as a sensitive metric for evaluation purposes and its measurement might be the reason for performance differences of text encoders when recovering different factors' contents. It is also mentioned by authors in Sec.3.4. but to further analysis. Based on the analysis of EM scores, there is not enough convincing evidence to suggest that the text encoder's ability to capture different factors varies. Therefore, the author should use different metrics for a thorough analysis, especially when Table 2 shows that the EM score may not support the conclusions of the paper. We also noticed in Appendix A that the author provided average results for BLEU-4 Score, which revealed smaller differences in text recoverability among various models. The author should further provide more detailed experimental results and analysis using different metrics.


3. In the Controlled-Imcaps experiment, there was no control over the differences in vision encoders among different VL-Models. This means that we cannot rule out the possibility that performance differences are caused by varying capabilities of vision encoders. To investigate this further, the author should consider more ablative experiment. Whether the varying capabilities of vision encoders could be a contributing factor to why text recoverability is not a sufficient condition for VL model performance is also a question. Additionally, this factor may also explain the weak performance of models in winoground.

**Reproducibility:**

4: Could mostly reproduce the results, but there may be some variation because of sample variance or minor variations in their interpretation of the protocol or method.

**Reviewer Confidence:**

3: Pretty sure, but there's a chance I missed something. Although I have a good feel for this area in general, I did not carefully check the paper's details, e.g., the math, experimental design, or novelty.

**Typos Grammar Style And Presentation Improvements:**

Table 2, along with other tables, is difficult to read. The author should consider restructuring the table. For example, they could highlight the data that supports their conclusions instead of making reviewers analyze a large number of tables. Additionally, it would be helpful to include average values in the table rather than only referencing them in the text without displaying them in the table.

---

> ### Author Rebuttal · Authors · 2023-08-29
>
> We thank all the reviewers for their detailed and constructive reviews. Thank you to Reviewer iU94 for pointing out our comprehensive analysis, insights, and the potential uses by the community for evaluation and informed model design.
>
> > **Detailed discussion of the results in Table 2:**
> * Thank you for pointing this out! Counter to most of our results, for which more details in the prompt make it harder to recover, the models indeed perform better at recovering adjective-noun pairs than single nouns (as seen in Table 2, “a cat” vs “an orange cat”). Looking at the errors, the models tend to output _more_ than is required for single nouns, i.e. they hallucinate incorrect information (e.g. “a tarantula” → “a tarantula in a hand”, “a schoolboy → “a schoolboy in school” in the case of CLIP ViT-B/32).
> * We will add a more detailed discussion to the final version of the paper that would target more specific phenomena such as the ones you pointed out, along with observations based on error analysis.
>
> > **T5 Decoder training:**
> * Thank you, we will certainly elaborate further on the T5 decoder training. The VL model text embedding is put through a linear transformation+layer normalization (L169-171) to convert it to a uniform embedding size fed into the T5 decoder.
>
> > **Why EM and not, e.g., BLEU?**
> * As suggested, we did consider text generation scores such as BLEU (Appendix Table 5). However, as discussed in L202-210, metrics such as BLEU and BERTScore are too lenient for our precise evaluation setup where one word can imply wildly different circumstances (e.g., “a cat on the left/right”). We think exact match is a fair evaluation metric for measuring information loss in a setup where models should be able to perfectly recover their inputs.
>
> > **Impact of Vision Encoder on ControlledImCaps results:**
> * We agree that an error on ControlledImCaps could be because of the vision encoder: in Sections 4.4-4.5 we discuss text recoverability as a necessary (but not sufficient) condition --- the insufficiency stems from the vision encoder, among other factors. We will add this discussion to the paper.
> * Regarding control over varying vision encoders, three of our four models actually use the same vision encoder architecture: CLIP ViT-B/32, negCLIP ViT-B/32 and RoBERTaCLIP ViT-B/32. The parameters of these vision encoders, however, are different, because they were trained in conjunction with their varying text encoders.
>
> Thank you again for your thoughtful comments, and your suggestions to improve the readability of our results. Please let us know if you have any further questions or comments, we would be happy to follow up!

---

### Meta-Review · Area_Chair_UmP7 · 2023-09-13

**Recommendation:** 5

**Metareview:**

This paper presents a thorough examination of the information loss within the language module of CLIP, aiming to understand its impact on language processing. To facilitate their investigation, the researchers introduce two novel datasets, namely CompPrompts and ControlledImCaps. The compelling analysis leads them to a finding: the text encoders in CLIP act as performance bottlenecks, potentially affecting the model's overall language capabilities.


This is a strong paper, in its observation, experiments and the benchmarks it provides.

---

### Decision · Program_Chairs · 2023-10-07

**Decision:**

Accept-Main

**Comment:**

This paper presents a thorough examination of the information loss within the language module of CLIP, aiming to understand its impact on language processing. To facilitate their investigation, the researchers introduce two novel datasets, namely CompPrompts and ControlledImCaps. The compelling analysis leads them to a finding: the text encoders in CLIP act as performance bottlenecks, potentially affecting the model's overall language capabilities.


This is a strong paper, in its observation, experiments and the benchmarks it provides.